# Identifiability of phenotypic adaptation from low-cell-count experiments and a stochastic model

Alexander P. Browning [1,2]*, Rebecca M. Crossley [2], Chiara Villa [3,4], Philip K. Maini [2], Adrianne L. Jenner[5], Tyler Cassidy[6], Sara Hamis[7]

**1** School of Mathematics and Statistics, University of Melbourne, Melbourne, Victoria, Australia, **2** Mathematical Institute, University of Oxford, Oxford, United Kingdom, **3** Sorbonne Université, CNRS, Université de Paris, Inria, Laboratoire Jacques-Louis Lions UMR, Paris, France, **4** Université Paris-Saclay, Inria, Centre Inria de Saclay, Palaiseau, France, **5** School of Mathematical Sciences, Queensland University of Technology, Brisbane, Queensland, Australia, **6** School of Mathematics, University of Leeds, Leeds, United Kingdom, **7** Department of Information Technology, Uppsala University, Uppsala, Sweden

* alex.browning@unimelb.edu.au

**Data availability statement:** Code used to produce the results is available at https://github.com/ap-browning/phenotypic_ heterogeneity_ibm.

## Abstract

Phenotypic plasticity contributes significantly to treatment failure in many cancers. Despite the increased prevalence of experimental studies that interrogate this phenomenon, there remains a lack of applicable quantitative tools to characterise data, and importantly to distinguish between resistance as a discrete phenotype and a continuous distribution of phenotypes. To address this, we develop a stochastic individual-based model of plastic phenotype adaptation through a continuously-structured phenotype space in low-cell-count proliferation assays. That our model corresponds probabilistically to common partial differential equation models of resistance allows us to formulate a likelihood that captures the intrinsic noise ubiquitous to such experiments. We apply our framework to assess the identifiability of key model parameters in several population-level data collection regimes; in particular, parameters relating to the adaptation velocity and cell-to-cell heterogeneity. Significantly, we find that cell-to-cell heterogeneity is practically non-identifiable from both cell count and proliferation marker data, implying that population-level behaviours may be well characterised by homogeneous ordinary differential equation models. Additionally, we demonstrate that population-level data are insufficient to distinguish resistance as a discrete phenotype from a continuous distribution of phenotypes. Our results inform the design of both future experiments and future quantitative analyses that probe phenotypic plasticity in cancer.

## Author summary

Many cancers adaptively and reversibly develop resistance to treatment, adding complexity to predictive model development and, by extension, treatment design. While so-called

**Funding:** APB thanks the Mathematical Institute, University of Oxford, for a Hooke Research Fellowship. SH was funded by Wenner-Gren Stiftelserna/the Wenner-Gren Foundations (WGF2022-0044) and the Kjell och Märta Beijer Foundation. RMC would like to thank the Engineering and Physical Sciences Research Council (EP/T517811/1) and the Oxford-Wolfson-Marriott scholarship at Wolfson College, University of Oxford (SFF2122-OWM-1091340) for funding. ALJ thanks the London Mathematical Society. CV is a Fellow of the Paris Region Fellowship Programme. This work was partially supported by a Heilbronn Institute for Mathematical Research Small Maths Grant to TC. The funders had no role in study design, data collection and analysis, decision to publish, or preparation of the manuscript.

**Competing interests:** The authors have declared that no competing interests exist.

drug challenge experiments are now commonly employed to interrogate phenotypic plasticity, there are very few quantitative tools available to interpret the biological data that arises. In particular, it remains unclear what is needed from drug challenge experiments in order to identify the phenotypic structure of a population that responds adaptively to treatment. In this work, we develop a new individual-level mathematical model of phenotypic plasticity in parallel with a structured model calibration process. Applying our framework to various existing and potential experimental designs reveals that experiments that yield only population-level data cannot distinguish between drug resistance as a distinct cell state, or drug resistance as a continuum of cell states. Consequentially, at the population-level, we demonstrate that common mathematical models that assume a set of distinct cell states can characterise the behaviour of cell populations that, in actuality, respond through a continuum of states. Importantly, our results shed light on both the mathematical models and experiments required to capture phenotypic plasticity in cancer.

## 1. Introduction

Phenotypic plasticity is widely acknowledged as a significant factor in the eventual failure in the treatment of many cancers [1–4]. Such short-term phenotypic adaptation arises in isogenic populations through epigenetics such that cells quickly manifest a reversible drug-tolerant phenotype when exposed to sufficiently high doses of a therapeutic drug [5,6]. Both experimental [6,7] and theoretical [8,9] studies have proposed adaptive therapy and the intermittent delivery of drug to overcome this phenomenon. Mathematical models, in particular, have been proposed to characterise this behaviour; interpret experimental studies of phenotypic adaptation; and to develop treatment schedules robust to resistance [10–16].

Complicating the characterisation of tumour-level plasticity within a given cancer are contrasting observations in the literature that resistance corresponds to a well-defined discrete phenotype [17–20] and to a continuous spectrum of phenotypes [21,22]. Indeed, many mathematical models of resistance describe a heterogeneous population comprising cells that are either firmly drug-sensitive or drug-resistant [18,23–25]. While mathematical models that capture continuous phenotype adaptation have been proposed [26–28] and are in fact well studied in the partial differential equation (PDE) literature [21,29], they remain largely unvalidated with experimental data.

Despite an increased prevalence of experimental studies that interrogate adaptive plasticity, there remains a lack of quantitative tools to calibrate models of phenotypic plasticity to experimental data. Thus key questions—such as the data requisite to identify the mechanisms behind adaptive plasticity, and the ability to distinguish between resistance as a discrete phenotype and a continuous distribution of phenotypes —remain unanswered. Mathematically, the question of whether model parameters can be estimated from experimental data is broadly referred to as *parameter identifiability* [30]. More specifically, if distinct parameter sets always lead to distinct model outputs (i.e., the parameter to output map is bijective), a model is said to be *structurally identifiable* [31]. In many cases, however, models may be structurally but not *practically* identifiable: that is, model parameters cannot be accurately estimated from a finite amount of noisy experimental data. It is this more pragmatic question, which relates directly to the experimental design required to reliably estimate parameters of interest, that we are primarily concerned with in this paper. Thus, from hereon we use the term *identifiability* to refer to practical identifiability. Issues relating to the identifiability of mechanisms behind adaptive plasticity from these models are likely to be further exacerbated

by other sources of cell-to-cell variability present in even isogenic cell populations [32], and by potential model misspecification.

We are motivated by a recent study of intermittent therapy of mutant melanoma cells by Kavran et al. [6], in which the authors provide genetic evidence for the presence of at least two (reversible) phenotype states: a drug-resistant phenotype and a drug-sensitive phenotype arising within a seven-day period of drug exposure and drug removal, respectively. From reported cell fold-change data from each phenotype, we have previously quantified a dose and phenotype dependent difference in net growth rate (Fig 1a); a characteristic of high interest in the context of the eventual development of treatments robust to adaptation [33]. Notably, Kavran et al. [6] provide compelling evidence for a continuous distribution of phenotypes present in the period between days 7 and 14 as the cells resensitise through observations of the cell-adhesion marker L1CAM (reproduced in Fig 1b), a protein well-known as a marker of the epithelial-to-mesenchymal transition and drug resistance in melanoma [34]. While

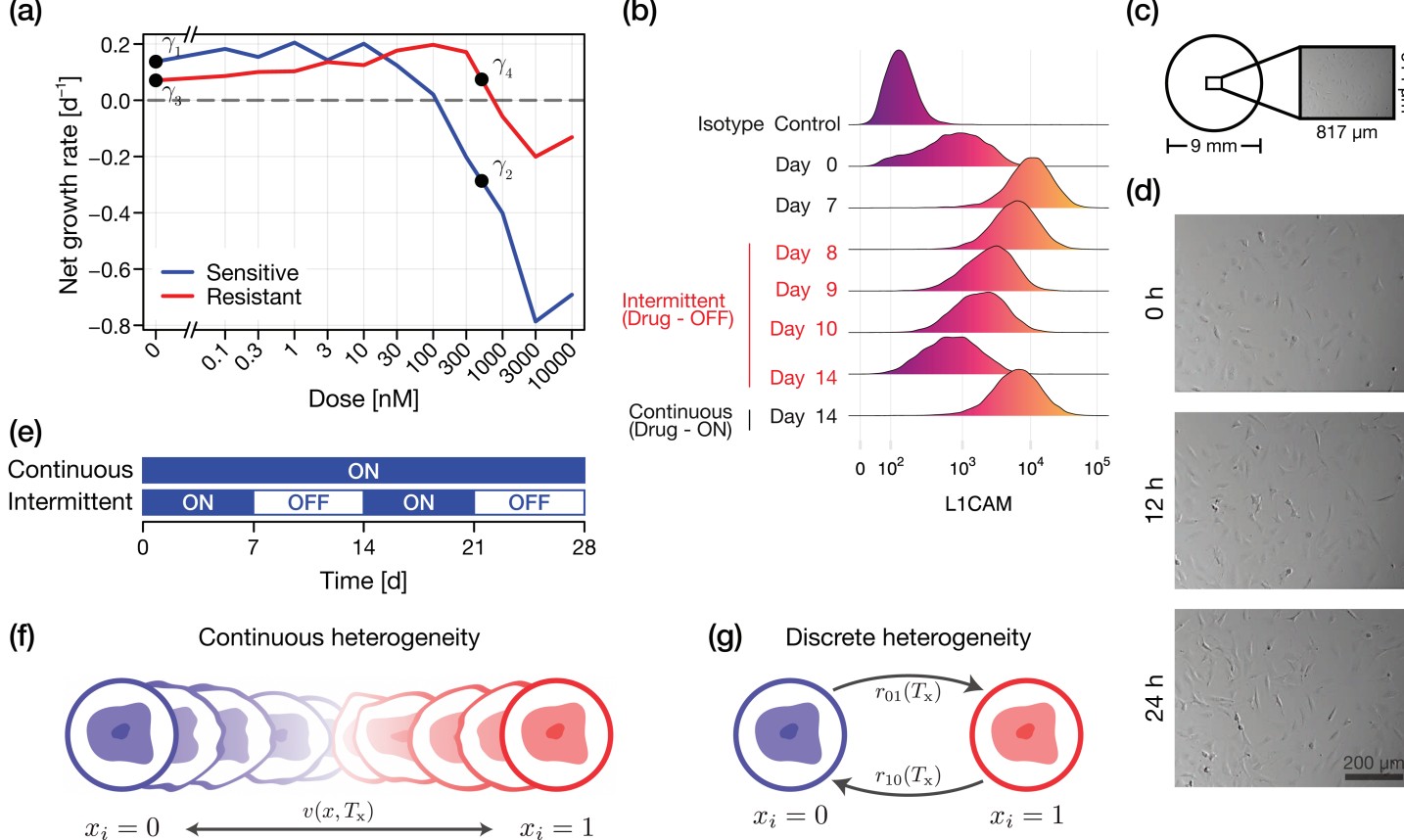

**Fig 1. Experimental data of phenotype adaptation.** An example suite of experimental data of phenotype adaptation. (a) Kavran et al. [6] expose WM239A melanoma cells to either a continuous treatment, or an intermittent treatment, as shown in (e). Net growth rate of cells calculated for various drug dose levels [33]. Cells that have been exposed to drug in the 7 days prior to measurement are classified as drug-resistant; cells that have not as drug-sensitive. (b) Phenotype characterised experimentally by the expression of L1CAM, a marker for cell adhesion. Day 14 intermittent data (i.e., cells that have not been exposed to drug between days 7 and 14) show a similar profile to day 0. Reprinted from [6] with permission from the author. (c) Schematic of a cell proliferation assay; cells grow on the substrate of 9 mm wells, and a central region is imaged at various time points. (d) Example suite of cell proliferation assay data; experiments conducted with a low density of 3T3 Fibroblast cells (reprinted from [35] under a CC-BY license and further analysed in [36]). (e) Cells are subject to either *continuous treatment*, in which a drug concentration is maintained, or to *intermittent treatment*, in which treatment alternates between 7-day periods of drug exposure and drug removal. (f,g) Schematics of continuous and discrete models of phenotypic heterogeneity, respectively (see text for details).

sequence and protein data provide qualitative insight into the adaptive dynamics, their link to cell growth rate is unlikely to be direct. We must, therefore, resort to using cell count data arising from proliferation assay experiments (Fig 1c) to quantify adaptive dynamics and the corresponding, possibly heterogeneous, net cellular growth rate.

To capture the stochasticity intrinsic to low-cell count experiments such as proliferation assays, we develop an individual-based model (IBM) of drug-based adaptation [27]. We build on the IBM of Hamis et al. [33] in a stochastic differential equation (SDE) framework to present a model in a continuous phenotypic space where a population of cells tend reversibly toward either a drug-sensitive state (mathematically, denoted by $x_i = 0$ where $x_i$ denotes the phenotype of cell $i$) or a drug-resistant state (denoted by $x_i = 1$). Changes in cellular phenotype are driven by two key mechanisms. First, deterministic drug-responsive movement described by a function of potentially unknown analytical form. Second, by a random diffusive process that induces cell-to-cell heterogeneity. For simplicity, all cells are otherwise statistically identical. The choice to work in an SDE framework means that our IBM corresponds precisely in a probabilistic sense to common PDE models of phenotypic adaptation [37].

Exploiting the tractability of the SDE and analogous PDE model, we build an inference framework that captures intrinsic noise in low-cell-count proliferation assay experiments without the pervasive, but often unjustified, assumption that experimental observations of cell count are subject to additive Gaussian noise. To do this, we derive and present a chemical master equation (CME) that describes the time-evolution of cell count, with which we construct a likelihood function [38]. We then perform inference and identifiability analysis under three data collection scenarios. First, we consider a suite of cell proliferation experiments for cells that are initially either resistant or sensitive and are exposed (or not) to a drug over a seven day period. Second, we consider a hypothetical scenario in which proliferation assays are observed continuously such that the time of cell-proliferation and cell-death events are directly observed. Lastly, we consider another hypothetical scenario in which a cell proliferation marker (i.e., L1CAM) correlates weakly, but linearly, with cell proliferation. All analysis is initially conducted in an idealised scenario where the functional form of the phenotype adaptation mechanism is correctly specified. We later relax this assumption and perform model selection.

Together, the data collection regimes we study establish the identifiability of individual model parameters and, more importantly, our ability to distinguish the phenotypic heterogeneity induced by random changes in phenotype from possible model misspecification. Given that the entrenched model of reversible phenotypic adaptation is that of two well-defined discrete phenotypes, we conclude our study by investigating whether such discrete heterogeneity can be distinguished, using cell count data alone, from a model comprising a continuous phenotypic space. As we are primarily interested in establishing the theoretical identifiability of model parameters and mechanisms, in the main text we focus our analysis on regimes where cell counts are extracted from images precisely; we relax this assumption in the supplementary material (S5 File) by investigating where identifiability is lost if only imprecise measurements can be made.

## 2. Mathematical methods

### 2.1. Individual-based model

We assume that individuals undergo a biased random walk in phenotype space, such that the phenotype of a cell $i$, denoted $x_i$, is given by

$$\mathrm{d}x_i = v(x_i, T_\mathrm{x})\,\mathrm{d}t + \beta\,\mathrm{d}W_i, \tag{1}$$

where $T_\mathrm{x} \in \{0, 1\}$ indicates whether a drug is present or not present, respectively; $v(x_i, T_\mathrm{x})$ describes the adaptation velocity; $\beta$ describes the magnitude of diffusive movement throughout the phenotype space, and $W_i$ is a Wiener process. We further assume that, for $\beta = 0$, the system has a stable steady state at $x_i = 0$ for $T_\mathrm{x} = 0$ (this is referred to as the sensitive state), and likewise at $x_i = 1$ for $T_\mathrm{x} = 1$ (referred to as the resistant state).

We assume that the net cellular growth rate is phenotype-dependent, modelled as a linear function of $x_i$ [33], parameterised as

$$\lambda(x_i, T_\mathrm{x}) = \begin{cases} \gamma_1 + (\gamma_3 - \gamma_1)x_i & \text{if } T_\mathrm{x} = 0, \\ \gamma_2 + (\gamma_4 - \gamma_2)x_i & \text{if } T_\mathrm{x} = 1, \end{cases} \tag{2}$$

as shown in Fig 1a. Provided that the growth rate is monotonic in $x_i$, the functional form of $\lambda$ is arbitrary since we could, in theory, rescale the phenotypic space in Eq (1) and thus equivalently the functional form of $v$. Furthermore, we follow [33], and assume that $\lambda(x_i, T_\mathrm{x}) < 0$ corresponds solely to net death (apoptosis or necrosis), and $\lambda(x_i, T_\mathrm{x}) > 0$ corresponds solely to net proliferation. We further assume that both proliferation and death events occur according to a Poisson process. Upon death, a cell is removed from the population. Upon proliferation, a cell is replicated such that daughter cells are created with an (initially) identical phenotype index to the parent.

While we focus on analysis of synthetic data, we choose biologically realistic parameters based upon analysis on the emergence of reversible resistance to the BRAF-inhibitor LGX818 in BRAF$^{\mathrm{V600E}}$-mutant melanoma cells [6,33]. The growth rate parameters are chosen to be $\gamma_1 = 0.15$, $\gamma_2 = -0.3$, $\gamma_3 = \gamma_4 = 0.1$ to approximately match the mean growth rate of sensitive and resistance cells under drug and no drug conditions (see dose-response curve in Fig 1a). Very little information is available regarding the adaptation dynamics through $v(x_i, T_\mathrm{x})$ and diffusivity $\beta$, other than the qualitative observation that cells move between drug-sensitive and drug-resistant states within a seven day window. We set

$$v(x_i, T_\mathrm{x}) = -\nu(x_i - T_\mathrm{x}) \tag{3}$$

with $\nu = 0.4$ such that $x_i$ is an Ornstein-Uhlenbeck process. We revisit this assumption with a more general form in Sect 3.4. Finally, we set $\beta = 0.05$ such that the stationary distribution of sensitive cells has a standard deviation of approximately 0.05. Implicit in our model is an assumption that the mechanisms behind drug-sensitisation and the reverse are identical. However, this need not be the case as we later exposit: it is sufficient to study identifiability in a single direction.

We set the initial condition in the model to a probabilistic representation of a spatially uniform low-cell count proliferation assay experiment; specifically, a cell proliferation assay conducted in a standard 9 mm well initialised with approximately 1000 cells (this is slightly larger than the initial population in [6]). The field-of-view of the imaged proliferation assay in Fig 1c–1d is 817×614 µm, and so each cell has probability $\rho = 817 \times 614/(4500^2\pi)$ of presenting in the field-of-view. The initial condition is thus set to $n_0 \sim \text{Binomial}(1000, \rho)$, corresponding to a mean initial cell count of approximately 7.9 per image.

In Fig 2a–2d, we simulate a set of synthetic cell proliferation assay experiments with our IBM under both continuous and intermittent treatments; the latter is defined as alternating 7-day periods of drug and no drug (Fig 1d). Results in Fig 2a, 2b highlight emergent isogenic heterogeneity due to white-noise driven fluctuations in the phenotype index. Results in Fig 2c,

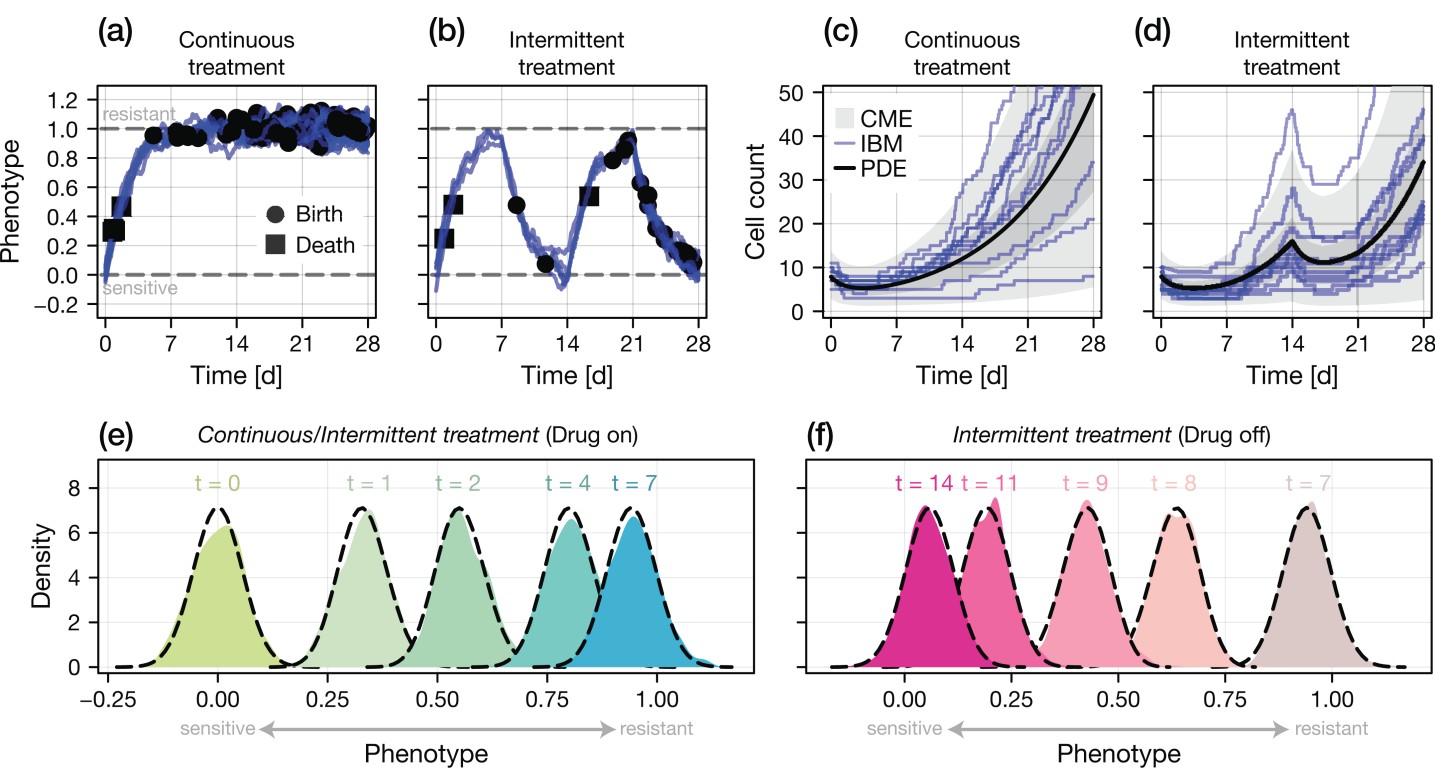

**Fig 2. Model comparison.** We compare realisations of the SDE-based IBM to the solution of both the corresponding Fokker-Planck PDE (Eq 4) and the CME (Eq 12) for the population size. (a–b) A single realisation of an IBM initiated with drug sensitive cells. The mean phenotype is zero in the equilibrium drug-sensitive state, and unity in the equilibrium drug-resistant state. Also shown are the timings of birth and death events. Treatment applied is (a) continuous; and (b) intermittent (Fig 1e). (c–d) Cell count observations from 10 realisations of the IBM (blue) under both (c) continuous and (d) intermittent treatment. Also shown is the expected population $\langle N \rangle(t)$ computed from a numerical solution of the PDE (black), and both a 50% and 95% credible region computed from a numerical solution of the CME (grey). A full comparison between the solution of the CME and the IBM is provided as supplementary material (S1 File). (e–f) Comparison between the phenotypic distribution computed empirically using an IBM initiated with 500 cells (coloured) and from the PDE (black dashed). Results in (e) show the phenotype distribution for both continuous and intermittent treatment for $0 \leq t \leq 7\,\mathrm{d}$ (in which both regimes are identical) and in (f) for intermittent treatment from $7 \leq t \leq 14\,\mathrm{d}$.

2d show high levels of stochasticity in cell count. Since the simulations are discrete, there is a non-zero probability of extinction as our model does not, in its standard formulation, consider migration into and out of the cell proliferation assay field-of-view (Fig 1c).

## 2.2. Partial differential equation model

We now define $u(x,t)$ as the density of cells with phenotype $x$ at time $t$, such that the dynamics of $u(x,t)$ are governed by the Fokker-Planck equation

$$\underbrace{\frac{\partial u(x,t)}{\partial t} + \frac{\partial}{\partial x}\Big(v(x, T_x(t))u(x,t)\Big) = \beta\frac{\partial^2 u(x,t)}{\partial x^2}}_{\text{Fokker-Planck equation}} + \underbrace{\lambda(x, T_x(t))u(x,t)}_{\text{Proliferation and Death}}, \quad (4)$$

subject to the usual set of no-flux and vanishing far-field boundary conditions [27,37,39].

Experiments are initiated with a sample of cells from a zero-net-growth stationary distribution for either a sensitive or resistant population. For the Ornstein-Uhlenbeck formulation of $v(x, T_x)$, this corresponds approximately to

$$x_i(0) \sim \mathcal{N}(0, \beta/\sqrt{2\nu}), \tag{5}$$

which we set as the initial condition in the model.

We denote by $\langle N \rangle(t)$ the expected cell count and by $p(x,t)$ probability density function (PDF), given by

$$\langle N \rangle(t) := \int_{\mathbb{R}} u(x,t) \, dx, \tag{6}$$

and

$$p(x,t) := \frac{u(x,t)}{\langle N \rangle(t)}, \tag{7}$$

respectively.

In Fig 2e–2f, we compare a finite-difference approximation to the PDE to a set of realisations of the IBM initialised with a large ($n_0 = 500$) number of initial cells. We remind the reader that we expect a close match (that converges as $n_0 \to \infty$), as the PDE is an exact probabilistic representation the IBM.

## 2.3. Chemical master equation

We now derive an approximate master equation for the time-evolution of the probability mass function for the cell count, defined as

$$q(n,t) := \mathbb{P}(N(t) = n). \tag{8}$$

We consider that

$$\begin{aligned}
q(n, t + \delta t) = q(n,t) &+ \mathbb{P}\big(\text{proliferation in } (t, t+\delta t) | N(t) = n-1\big) q(n-1, t) \\
&+ \mathbb{P}\big(\text{death in } (t, t+\delta t) | N(t) = n+1\big) q(n+1, t) \\
&- \mathbb{P}\big(\text{proliferation in } (t, t+\delta t) | N(t) = n\big) q(n, t) \\
&- \mathbb{P}\big(\text{death in } (t, t+\delta t) | N(t) = n\big) q(n, t).
\end{aligned} \tag{9}$$

Note that we can also include terms in the above that explicitly capture migration into and out of the field of view. Generally, however, we would expect these to vanish if we assume that the assay as a whole is sufficiently homogeneous such that migration out of the window occurs at the same rate as migration into the window (i.e., periodic boundary conditions).

To make progress, we assume that the phenotypic states of cells are independent. While not strictly true for very high proliferation and death rates (since cells inherit their phenotype from a parent), this is appropriate for the range of growth rates we observe (Fig 1a). Under these assumptions, the per-capita instantaneous proliferation and death rates are given by

$$r_{\text{prol}}(t) = \int_{\mathbb{R}} p(x,t) \max(0, \lambda(x,t)) \, dx, \tag{10}$$

and

$$r_{\text{death}}(t) = -\int_{\mathbb{R}} p(x,t) \min(0, \lambda(x,t)) \, dx, \tag{11}$$

respectively, where $p(x,t)$ is governed by the PDE (Eqs 4 and 7).

For $\delta t$ sufficiently small, we can consider a Taylor expansion of the exact Poisson probability to obtain an asymptotic expression for the event probabilities in Eq (9). These are given by

$$\mathbb{P}\big(\text{proliferation in } (t, t+\delta t)|N(t) = n\big) \sim n r_{\text{prol}}(t)\delta t,$$

and

$$\mathbb{P}\big(\text{death in } (t, t+\delta t)|N(t) = n\big) \sim n r_{\text{death}}(t)\delta t.$$

Substituting into the difference equation (Eq 9) and taking $\delta t \to 0$, we arrive at the CME

$$\begin{aligned}
\frac{\mathrm{d}q(n,t)}{\mathrm{d}t} = {}& (n-1)r_{\text{prol}}(t)q(n-1,t) \\
& - n\big(r_{\text{prol}}(t) + r_{\text{death}}(t)\big)q(n,t) \\
& + (n+1)r_{\text{death}}(t)q(n+1),
\end{aligned} \tag{12}$$

subject to absorbing boundaries such that $q(n) = 0$ for $n < 0$.

In Fig 2c–2d, we compare the solution of the CME to realisations of the IBM, showing that the CME captures both the average and variance of the cell count. A more detailed comparison is provided in S1 File.

## 2.4. Likelihood-based inference

We take a Bayesian approach to parameter estimation and identifiability analysis and apply the CME (Eq 12) to construct a likelihood for cell count data reported from proliferation assays. The advantage of this approach, compared to a more standard approach that considers an average cell count subject to additive Gaussian noise, is that we account directly for the stochasticity intrinsic to the proliferation death process. As we are primarily interested in the identifiability of model parameters, we assume that all cell counts are exact. In the supplementary material (S5 File), we investigate identifiability in the case that experimental observations are potentially subject to miscounting.

Experiments are conducted for $t$ days, at the conclusion of which a cell count observation is taken. We denote by $n_k^{(t,T_{\text{x}},\mathcal{P})}$ a cell count taken from the $k$th replication of an experiment terminated at time $t$, conducted entirely with ($T_{\text{x}} = 1$) or without ($T_{\text{x}} = 0$) drug, using an initial population of sensitive (denoted $\mathcal{P} = 0$) or resistant (denoted $\mathcal{P} = 1$) cells, and denote by $\mathcal{D}$ the complete set of data. Further denoting the solution to the CME with conditions $(\mathcal{P}, T_{\text{x}})$ and parameter values $\boldsymbol{\theta}$ by $q_{(\mathcal{P}, T_{\text{x}})}(n, t; \boldsymbol{\theta})$, the log-likelihood is given by

$$\ell_{\mathcal{D}}(\boldsymbol{\theta}) = \sum_{(\mathcal{P}, T_{\text{x}})} \sum_t \sum_k \log q_{(\mathcal{P}, T_{\text{x}})}\big(n_k^{(t,T_{\text{x}},\mathcal{P})}, t; \boldsymbol{\theta}\big). \tag{13}$$

Here, the summation is taken over all experimental conditions, all time points, and all experimental replicates. Note that we have assumed that cell count observations are independent between time points; effectively assuming that measurements are taken at the termination of an experiment and not as a time-series. While our approach could be trivially extended to account for time-series data, this would add significant computational cost by potentially requiring a numerical solution to the CME for each individual observation. While we focus our results on inference using cell count data, we also consider log-likelihood functions constructed for two other data types: event timing data (i.e., the exact time of proliferation or cell death observed from temporal data) and from a cell proliferation marker that may linearly correlate with the net growth rate.

Following the construction of the log-likelihood function, we can either take a frequentist approach and find the maximum likelihood estimate (MLE), or apply a Bayesian approach to quantify identifiability and parameter uncertainty. While unusual to consider both approaches, we do so in this work as the former is advantageous as it allows us to perform model selection using frequentist hypothesis tests [40].

For the latter, we assume that knowledge about model parameters is initially encoded in a *prior distribution*, $p(\theta)$. We choose $p(\theta)$ to be independent uniform over a sufficiently wide range of parameter magnitudes (full details are given in S1 File). This choice also ensures that the maximum *a posteriori* estimate (MAP) corresponds to the MLE. Following a set of observations, denoted by $\mathcal{D}$, arising from cell proliferation assay measurements, or otherwise, we update our knowledge about the model parameters using the relevant likelihood denoted $\ell_{\mathcal{D}}$ to obtain the *posterior distribution*, given by

$$p(\theta|\mathcal{D}) \propto \exp(\ell_{\mathcal{D}}(\theta))p(\theta). \tag{14}$$

When applying the Bayesian approach, we sample from the posterior using the adaptive scaling within adaptive Metropolis Markov-Chain Monte Carlo algorithm implemented by [41] in `AdaptiveMCMC.jl` with 10,000 iterations. To obtain MLEs we apply the DIRECT global search algorithm implemented in NLopt for Julia [42] to the likelihood function. Similarly, for MAPs we apply the same algorithm to the posterior density function. As we are primarily interested in parameter identifiability, which relates to whether the likelihood is flat in the vicinity of either the "true" or best fitting parameter values, for simplicity we initiated each chain using the "true" set of parameter values that are used to generate the synthetic data.

## 3. Results

### 3.1. Phenotypic heterogeneity is poorly identified from cell count data

We begin our analysis by considering a suite of synthetic cell proliferation assays conducted within a seven day period (specifically, a set of assays that terminate at $t$ = 1, 3, 5, and 7 d). For each termination time, we conduct a set of four experiments: with or without drug and initiated with either a population of fully sensitive or resistant cells. We devote two 96-well plates to each termination time, such that the sample size for each condition is $M$ = 48. The duration is chosen based on the observation that the population adapts or resensitises within a seven day interval [6] (in S2 File, we consider a variety of termination time sets).

Applying the CME-based Bayesian inference procedure reveals that all growth rate parameters are practically identifiable. The results in Fig 3a–3b show how model predictions produced at the MAP align with synthetic cell count data observations. Furthermore, results in Fig 3c show that the adaptation speed parameter, $\nu$, is identifiable. However, we see from results in Fig 3d that the diffusion parameter $\beta$, which corresponds to the variance in the phenotype variable $x$ within an adapting population, is only *one-sided identifiable*: we can establish an upper bound, but no lower bound. In the supplementary material, we show this to also be the case if temporally correlated cell count observations are made (S6 File). The parameter is, however, *structurally identifiable*: we show this in the supplementary material (S4 File) using a significantly larger ($M$ = 768) data set, however the parameter becomes again non-identifiable when imprecise cell-count observations are made (S5 File). Thus, from cell count data alone, we expect that models with a phenotypic heterogeneity (i.e., models with a random component to phenotype changes) to be indistinguishable from models with deterministic adaptation (the $\beta$ = 0 scenario).

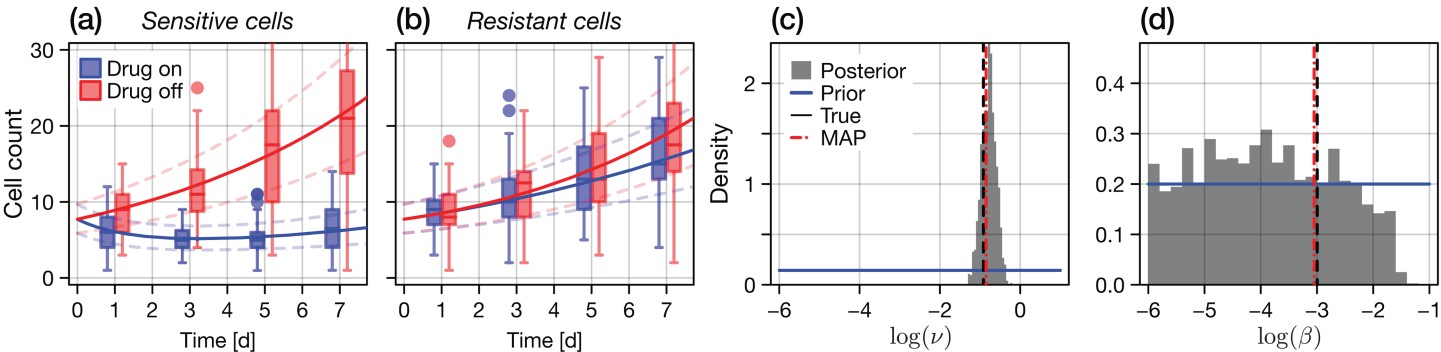

**Fig 3. Proliferation assay inference.** We perform Bayesian inference on a set of synthetic cell proliferation assay data using the CME as a likelihood. Independent cell count observations ($M$ = 48 replicates per condition) are collected from experiments conducted with fully sensitive or fully resistant cells, with and without drug, and terminated at $t = \{1\,\mathrm{d}, 3\,\mathrm{d}, 5\,\mathrm{d}, 7\,\mathrm{d}\}$. (a–b) Synthetic proliferation assay cell count data (box plots), the chemical master equation predicted median cell count at the MAP (solid lines), and the model predicted first and third quantiles (dashed semi-transparent lines). (c–d) Posterior distributions for the logarithms of $\nu$, the adaptation speed, and $\beta$, the diffusivity. Shown also is the uniform prior (blue), the true value (black dashed), and the MAP (red dashed). While the adaptation speed is identifiable (as are all other parameters; see S2 File), the diffusivity is only one-sided identifiable; the model cannot be distinguished from that with purely deterministic adaptation (i.e., no heterogeneity).

To investigate the identifiability of $\beta$ further, we recall that the phenotype distribution, $p(x,t)$, affects overall cell count dynamics only indirectly. Specifically, cell proliferation and death is governed at the population-level by the overall proliferation and death rates, given by Eqs (10) and (11). For $\beta = 0$, $p(x,t)$ tends to a degenerate distribution such that $r_{\mathrm{prol}}(t) = \lambda(\bar{x}, t)$, where $\bar{x}$ is the mean phenotype (in the case of homogeneity, the only phenotype). Following from our assumption that a positive net growth rate corresponds solely to proliferation, the most obvious consequence of the $\beta = 0$ parameter regime is that proliferation and death cannot occur simultaneously: thus, we expect a sharp proliferation-death transition at $r_{\mathrm{prol}}(t) = 0$ as the population switches between death and proliferation events, depending on the presence of drug and the mean phenotype. In contrast, the transition at $r_{\mathrm{prol}}(t) = 0$ will be diffuse for non-zero $\beta$. In Fig 4, we compare the event rates for various values of $\beta$. Clearly, aside from minor differences at the proliferation-death transition, rate curves are visually indistinguishable for decreasing values of $\beta$. For large $\beta$, which has very little or no posterior mass (see Fig 3d), the proliferation rate curve becomes distinguishable.

## 3.2. Phenotype heterogeneity is identifiable from event-timing data

Under the current model formulation, in which heterogeneity is driven solely by diffusion through the phenotypic space, it is only in the regime where $\beta > 0$ that we will ever see proliferation and death events occur simultaneously. Thus, in the constraints of our model formulation, we expect to be able to more precisely identify $\beta$ if we observe the precise timings of cellular proliferation and death events from, for example, live cell imaging.

We therefore investigate a hypothetical scenario where we have access to noise-free event timings from a set of proliferation assays that are initiated with a total of 10,000 cells. Without loss of generality, for the rest of the study we focus only on adaptation in the forward direction (i.e., from drug-sensitive to drug-resistant), since an analogous analysis could be conducted in the reverse direction. A log-likelihood function can be constructed by discretising the resultant Poisson process such that the number of proliferation and death events

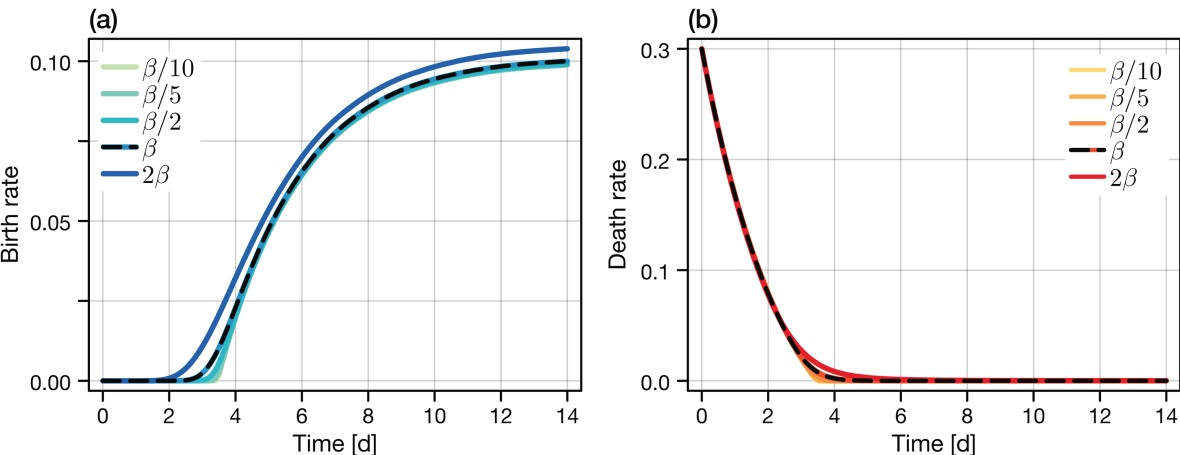

**Fig 4. Practical non-identifiability from cell count data.** Practical non-identifiability of the diffusion parameter $\beta$ (corresponding to a measure of the heterogeneity), seen through differences in the total expected proliferation and death rate functions, $r_{prol}(t)$ and $r_{death}(t)$, respectively. All other parameters are fixed at their true values. Clear differences are seen in the proliferation rate between $\beta$ and a proliferation rate constructed where $\beta \mapsto 2\beta$; we have seen previously that $\beta$ is one-sided identifiable. However, reducing $\beta$ shows (visually) very minor differences between the proliferation and death rate functions as both tend to the deterministic limit (in this case, corresponding to exponential decay from the negative maximum death rate through to the maximum proliferation rate). In the absence of heterogeneity (i.e., for $\beta \to 0$), proliferation and death events cannot occur simultaneously in a population.

occurring in the interval $(t, t+\delta t)$, denoted $E_{prol}^{(\delta t)}(t)$ and $E_{death}^{(\delta t)}(t)$, respectively, are distributed according to

$$E_{prol}^{(\delta t)}(t) \sim \text{Poisson}\big(r_{prol}(t)N(t)\delta t\big), \qquad E_{death}^{(\delta t)}(t) \sim \text{Poisson}\big(r_{death}(t)N(t)\delta t\big), \qquad (15)$$

where $N(t)$ is the (observed) cell population at time $t$. Under the well-mixed phenotype assumption for which the CME applies, Eq (15) is exact as $\delta t \to 0$. We choose $\delta t = 0.035$ d such that the seven-day experiment is subdivided into 200 observation intervals (as a consequence, one could also consider event-timing data that is not exact, but accurate to intervals of width $\delta t$ that correspond to a finite imaging frequency). As the intervals are non-overlapping, the observed number of birth and death events within each interval are statistically independent, and the log-likelihood is given simply through the probability mass function for the Poisson distribution in each interval.

The synthetic data set is shown in Fig 5a, along with an estimate for the instantaneous event rate constructed using a moving average. Visually, heterogeneity can be detected through the transition from primarily cell proliferation to primarily cell death. That is, for $\beta = 0$ the homogeneous population of cells will exclusively either proliferate or die, but not both. We proceed to perform inference on this synthetic data set using the Poisson likelihood, with the posterior shown for $\beta$ in Fig 5b (all other relevant parameters remain identifiable). Clearly, heterogeneity is now identifiable; estimates of $\beta$ can be drawn precisely.

## 3.3. Phenotype heterogeneity is not identifiable from proliferation marker data

Our study is in part motivated by Kavran et al. [6] who provide compelling evidence for a continuous transition from a sensitive to resistant state through the cell-adhesion marker L1CAM. Such data are difficult to interpret directly due to uncertainty in the precise link

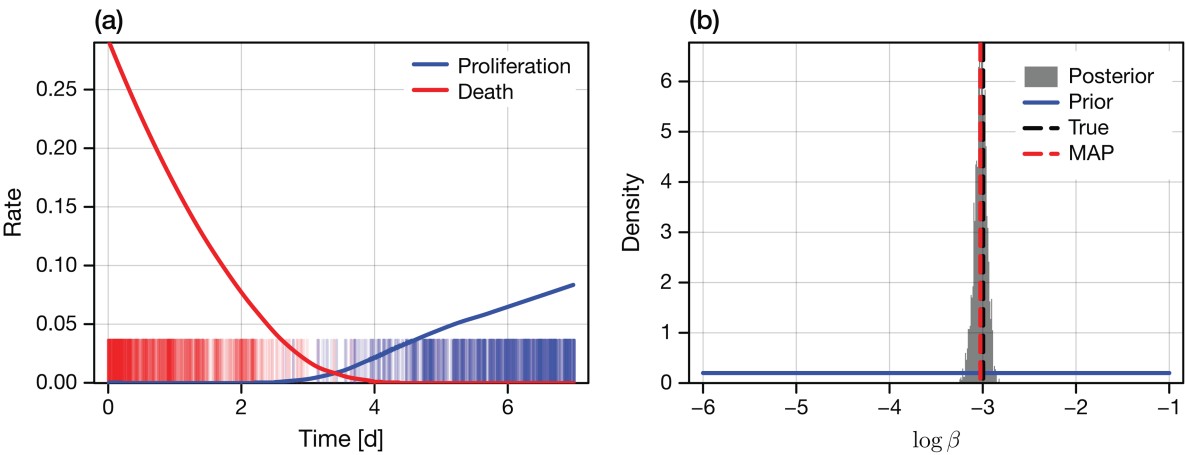

**Fig 5. Identifiability of heterogeneity from event timing data.** We generate a synthetic data set from an experiment (or set of experiments) that are initiated with a total of 10,000 cells that are under continuous treatment. The exact event timings (i.e., time of cell proliferation, and time of cell death) are recorded and used for inference. (a) Synthetic event timing data. Shown is a rug plot of a sample of 500 each of proliferation and death events, and a local regression (LOESS) of the observed proliferation and death rate. (b) Posterior distribution for $\log(\beta)$, previously non-identifiable, constructed using a Poisson likelihood for the exact timing data. Shown also is the uniform prior (blue), the true value (black dashed), and the MAP (red dashed).

between the net growth rate and the expected marker expression and the resultant flow cytometry measurement. Challenges aside, we now consider identifiability of $\beta$ in the case that the measured marker expression correlates linearly with the proliferation rate (and effectively, since the link between the net growth rate and phenotype index is also linear, the phenotype index).

We assume that the observed marker expression for cell $i$, denoted by $M_i$, is given by

$$M_i \sim x_i + \varepsilon \tag{16}$$

where $\varepsilon$ is independent of $x_i$. We consider both that $\varepsilon$ is normally distributed with zero mean and unknown standard deviation $\sigma$, and a scenario where the shape of $\varepsilon$ is additionally unknown such that $\varepsilon$ is given by a translated Gamma distribution with zero mean, unknown standard deviation, and unknown skewness $\omega$ (this distribution becomes normal as $\omega \to 0$) [43]. By convoluting the distribution of $\varepsilon$ with that for $x$, we can construct an exact log-likelihood for a set of marker data. In Fig 6a, we show the resultant (weak) linear correlation between phenotype index and marker measurement.

We fix all other mechanistic parameters, which we previously established to be identifiable from cell count data, at the corresponding true values. We then consider a synthetic data set in which marker data is taken from a set of proliferation assays terminated at $t = \{1\,\mathrm{d}, 3\,\mathrm{d}, 5\,\mathrm{d}, 7\,\mathrm{d}\}$. Results in Fig 6b show samples from the joint posterior distribution for $\log(\sigma)$ and $\log(\beta)$ in the case that $\omega = 0$. In both the case where the marker error shape is known ($\omega = 0$) and unknown, we are unable to place a lower bound on $\beta$. Furthermore, the shape of the posterior in Fig 6b indicates that, even if we had knowledge of $\sigma$, $\beta$ would remain only one-sided identifiable. We conclude that, from a marker that does not correlate perfectly with growth rate, heterogeneity in the proliferation rate is indistinguishable from marker noise.

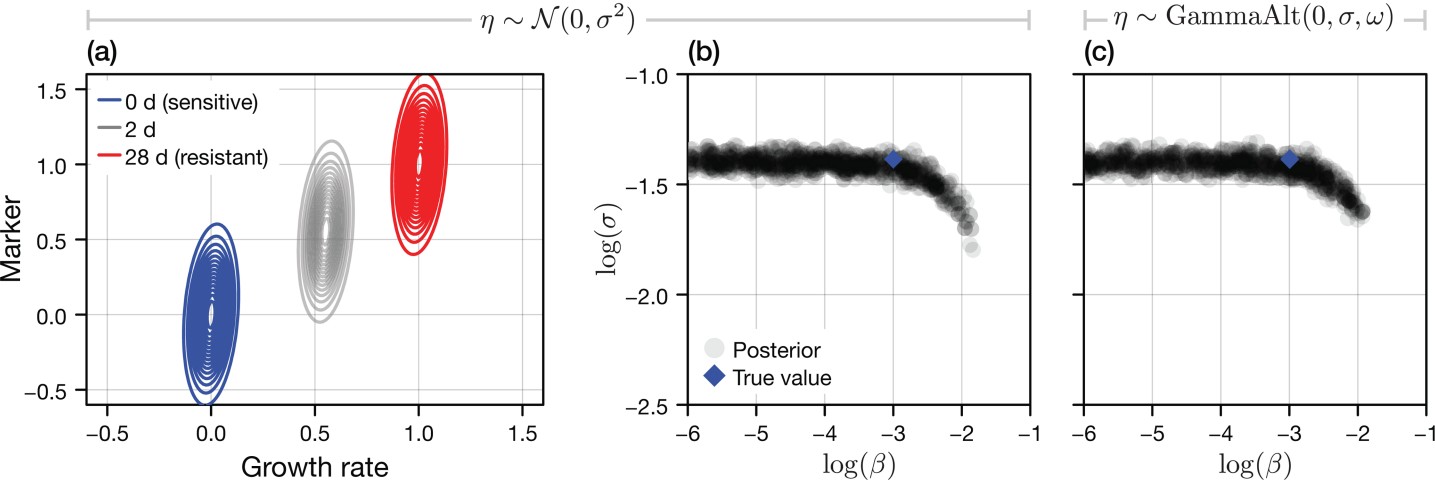

**Fig 6. Identifiability of heterogeneity from noisy marker data.** We generate a synthetic data set comprising noisy measurements of the phenotype state of each cell using a hypothetical marker for cell proliferation (i.e., L1CAM). (a) Measurements are normalised such that the mean of fully sensitive cells is approximately zero, and that of fully resistant cells (which arise in the limit as $t \to \infty$) is approximately unity. The marker is assumed to weakly linearly correlate with growth rate (and hence, the phenotype index); we model this by a measurement noise process that is normally distributed with variance $\sigma^2$. (b) We perform Bayesian inference on a dataset generated from cell proliferation assays with fully sensitive cells, exposed to drug continuously, with independent measurements taken at $t = \{1\,\mathrm{d}, 3\,\mathrm{d}, 5\,\mathrm{d}, 7\,\mathrm{d}\}$ ($M = 48$ replicates per measurement time). All other parameters, identifiable from cell count data, are fixed, and $\sigma$ and $\beta$ are estimated, with the joint posterior (grey discs) shown alongside the true value (blue diamond). (c) We repeat the analysis in the case that the shape (skewness, quantified by $\omega$) of the measurement noise distribution is additionally unknown.

## 3.4. Model selection and misspecification for cell-count data

We have made two significant observations thus far: first, that the $\beta > 0$ regime is indistinguishable from the $\beta = 0$ regime from cell count data; and second, that all other model parameters are identifiable given a correctly specified model. As a consequence of the first observation, we perform all remaining analysis using what we term the "homogeneous continuous model": an ordinary differential equation (ODE) model given by Eq (1) with $\beta = 0$. Therefore, all cells carry the same phenotype, denoted now by $x(t)$. Our goal now is assess whether we can not only identify model parameters, but also the functional form of the adaptation velocity $v(x, T_\mathrm{x})$ (without loss of generality in the case that $T_\mathrm{x} = 1$ such that $v(x, 1) = u(x)$).

We consider a relatively general functional form for $u(x)$, given by

$$u(x) = a\,\mathrm{sgn}(1 - x) + (1 - x)(b + cx + dx^2), \tag{17}$$

where $\mathrm{sgn}(x)$ is the sign function. This form of $u(x)$ allows choices of increasing complexity to be recovered by setting parameters to zero. As before, we consider a set of synthetic cell proliferation assays conducted with drug sensitive cells under continuous treatment and terminated at $t = \{1\,\mathrm{d}, 3\,\mathrm{d}, 5\,\mathrm{d}, 7\,\mathrm{d}\}$ ($M = 48$ per condition). The true model (Eq 3) is recovered by setting $a = c = d = 0$. We can recover a variety of velocity models using the functional form given by Eq (17), including for $b = c = d = 0$ the constant adaptation presented in our previous work [33]. As the growth rate parameters for the drug-on experiment, $\gamma_2$ and $\gamma_4$, were found to be identifiable (and can be established by conducting drug-off and drug-on experiments with sensitive and resistant cells, respectively) we fix each to their corresponding true value.

We perform model selection using the frequentist likelihood ratio test (equivalent to profile likelihood). For example, to test whether $a = 0$, we compare the likelihood at the MLE

(equivalently, the MAP) where we fix $a = 0$ to that for the model where all parameters in Eq (17) are non-zero, denoted by $\hat{\theta}$. Fig 7a shows the resultant set of log-likelihoods, translated such that $\ell(\hat{\theta}) = 0$. From the likelihood ratio test [40], we can construct a threshold based on a 95% confidence interval outside of which we reject a null hypothesis that the parameter set, i.e., $[a]$ is equal to zero.

Results in Fig 7a show that any individual parameter can be set to zero. Furthermore, any pair of parameters can be set to zero *except a and b* simultaneously. Finally, only the parameter triples that do not contain *both a and b* can be set to zero. If the goal was to identify a single model, one would use an information criterion [40] (or similar) to penalise differences in log-likelihood by the dimensionality of the non-zero parameter set; in our case, we expect a model where only one of $a$ or $b$ is non-zero as the most parsimonious.

Our analysis has identified a family of possible adaptation velocity functions, given by the MLE for each combination for which the relative log-likelihood in Fig 7a is above the corresponding threshold. In Fig 7b we compare the identified adaptation velocities for the true model ($b$ non-zero) to the full model (no non-zero parameters) and a model where only $a$ is non-zero. Clearly, there remains large uncertainty as to the functional form of $u(x)$ throughout the phenotype space. Results in Fig 7c, however, demonstrate why these differences do not manifest in statistically different cell count observations: while $u(x)$ varies significantly, the possible paths for $x(t)$ are similar.

## 3.5. Continuous and discrete-binary heterogeneity may be indistinguishable

Arguably the standard model of plasticity describes a drug-dependent switch between two discrete phenotypes: sensitive and resistant. Such an analogue of our model is

$$X_0 \underset{r_{10}(d)}{\overset{r_{01}(d)}{\rightleftharpoons}} X_1, \tag{18}$$

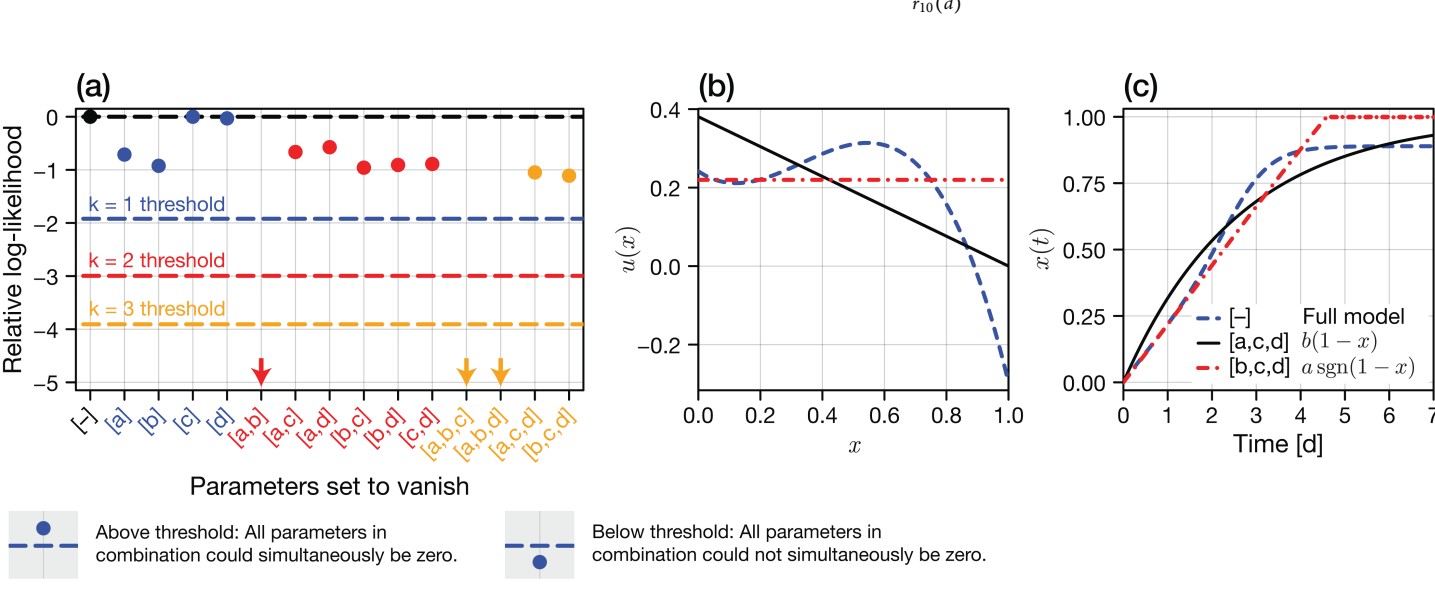

**Fig 7. Model selection and misspecification.** We perform inference and model selection on a general adaptation velocity function of form given by Eq (17). The true model corresponds to $b = 0.4$ and $a = c = d = 0$ (i.e., the combination $[a, c, d]$). (a) Results from a likelihood ratio test where the null hypothesis in each column is that the stated parameter combination $[\cdot]$ is zero. Relative log-likelihood values below the relevant threshold (colours correspond to different dimensionalities) indicate that the null hypothesis can be rejected at the level of a 95% confidence interval. Arrows indicate that observed statistics are below the plotted region. (b,c) Identified possible adaptation velocities and phenotype transitions respectively.

where sensitive cells, $X_0$, have net growth rate $\tilde{\lambda}_0(d)$ dependent on the drug concentration $d$, and resistant cells, $X_1$, have net growth rate $\tilde{\lambda}_1(d)$ (Fig 1g). We assume that $r_{10}(d)$ and $r_{01}(d)$ are also drug-dependent.

As Eq (18) is linear, the mean cell count in each subpopulation, denoted by $n_0(t)$ and $n_1(t)$, is given by

$$\begin{aligned}
\frac{dn_0}{dt} &= \tilde{\lambda}_0(d)n_0 - r_{01}(d)n_0 + r_{10}(d)n_1, \\
\frac{dn_1}{dt} &= \tilde{\lambda}_1(d)n_1 + r_{01}(d)n_0 - r_{10}(d)n_1.
\end{aligned}$$

(19)

To draw a correspondence to the continuous model, we consider now the mean $\tilde{x}(t) := n_1(t)/(n_0(t) + n_1(t))$, which we expect to correspond with $x(t)$ in the continuous model (although not exactly, as in general at equilibrium $\tilde{x}(t) \neq 1$). The dynamics of $\tilde{x}(t)$ are governed by

$$\begin{aligned}
\frac{d\tilde{x}}{dt} &= r_{01}(d) + \left(\tilde{\lambda}_1(d) - \tilde{\lambda}_0(d) - r_{01}(d) - r_{10}(d)\right)\tilde{x} + \left(\tilde{\lambda}_0(d) - \tilde{\lambda}_1(d)\right)\tilde{x}^2 \\
&= A(d) + B(d)\tilde{x} + C(d)\tilde{x}^2.
\end{aligned}$$

(20)

Thus, we expect the average cell count in the discrete model to correspond exactly to the average cell count in a continuous model with a quadratic and drug-dependent adaptation velocity. We cannot make an equivalent statement for higher order moments, however we can define an exact CME for the evolution of the joint density $\tilde{q}(n_1, n_2, t) := \mathbb{P}(N_1(t) = n_1, N_2(t) = n_2)$ and hence the probability mass $\tilde{q}(n, t) := \mathbb{P}(N_1(t) + N_2(t) = n)$ in the discrete model (S3 File).

For a given set of discrete model parameters, we compute a rescaled velocity function and set of continuous model net growth rates such that both models have equivalent initial and fully adapted net growth rates. In Fig 8a, we demonstrate under continuous application of the drug that the mean cell counts are identical between models. Therefore, from average cell count data, and by extension large-cell-count proliferation assays, we cannot distinguish a discrete model from a continuous model with quadratic adaptation velocity. Results in Fig 8b–8d demonstrate (subtle) differences in higher-order moments and the mass function for each model. We conclude, therefore, that within our modelling framework it may be possible to distinguish between the discrete and continuous models using higher order moments in low-cell-count proliferation assays; however this is unlikely to be the case if only imprecise cell count observations are available. Provided that the adaptation velocity is drug-dependent (i.e., cells sensitise at a rate different to that at which they develop resistance), these findings also apply for so-called intermittent treatment [33,44]. In Fig 8e–8g, we demonstrate that this equivalence between the discrete and continuous model holds for a variety of different treatment schedules.

## 4. Discussion and conclusion

Phenotypic plasticity and the rapid adaptation of cells upon the application of treatment are widely recognised as a significant factor in the failure of many anti-cancer treatments [45]. Complicating a comprehensive characterisation of phenotypic plasticity is a lack of consensus as to whether adaptation occurs between a set of well-defined discrete cell states or across a continuous spectrum of phenotypes. While both hypotheses are associated with mature subsets of the mathematical modelling literature, there remains—particularly for the latter—a

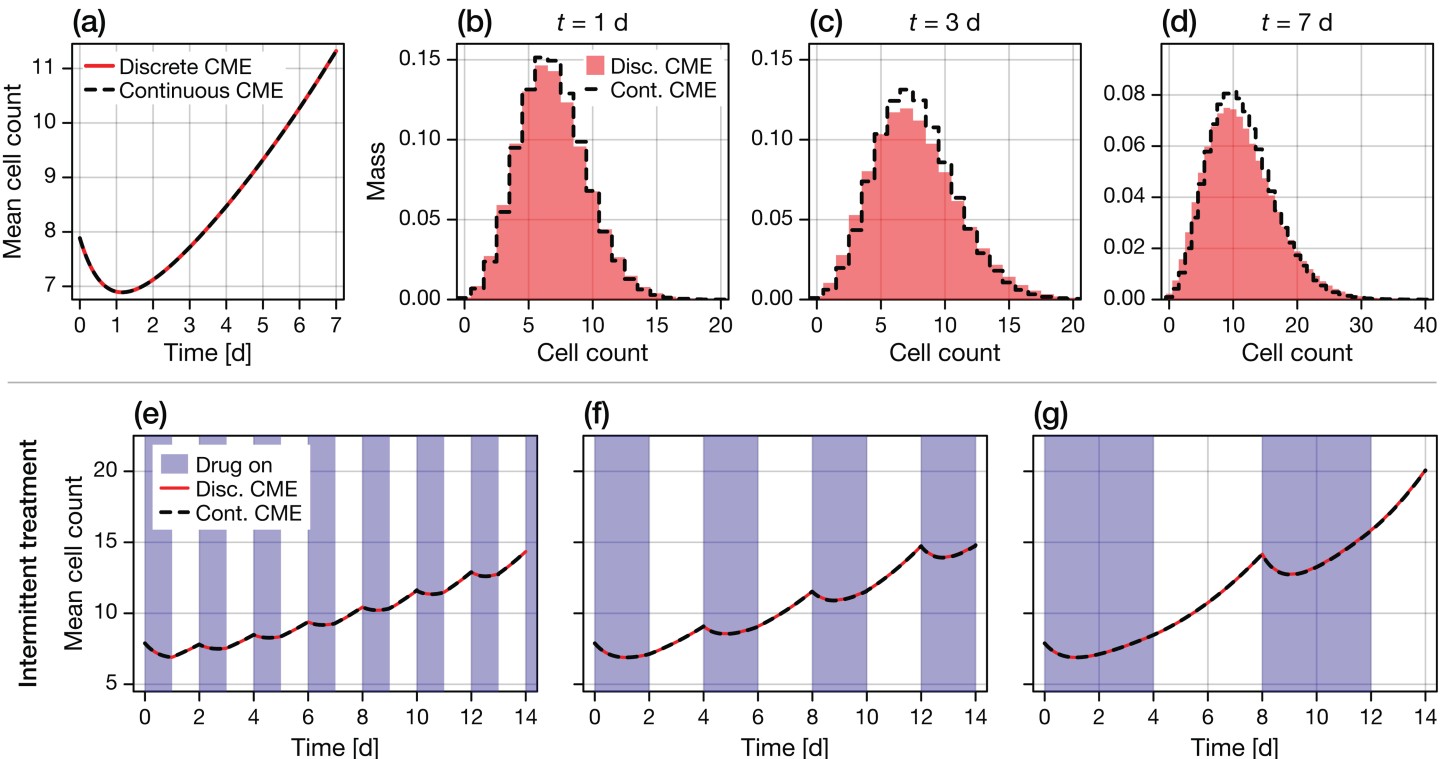

**Fig 8. Continuous and discrete-binary models are only distinguishable from higher order moments.** We compare the solutions of the CME for a discrete-binary model and a continuous model with quadratic adaptation velocity under (a–d) continuous application of a drug, and (e–g) intermittent application of a drug. In (e–f), blue regions indicate time periods during which the drug is present, white regions indicate time periods when the drug is removed. (a,e,f,g) Exact correspondence in the mean cell count for each model; the regimes are non-identifiable. (b–d) Solution to the complete CME under continuous drug treatment at various time points, showing higher-order differences between the models. Discrete model parameters are give by $r_{01}(1) = 1$, $r_{10}(1) = 0.01$, $\hat{\lambda}_0(1) = -0.3$, $\hat{\lambda}_1(1) = 0.1$ when the drug is applied (i.e., $d = 1$) and $r_{01}(0) = 0.02$, $r_{10}(0) = 0.5$, $\hat{\lambda}_0(0) = 0.15$, $\hat{\lambda}_1(0) = 0.1$ when the drug is removed (i.e., $d = 0$).

dearth of statistical methodology to parameterise such models. Indeed, key questions relating to the identifiability of adaptation mechanisms and the within-population heterogeneity arising out of random diffusive phenotypic changes, the ability of practitioners to distinguish between discrete and continuous adaptation, and the experimental design requirements to parameterise models, remain unanswered.

Our most significant result is that we are unable to detect heterogeneity arising from random phenotypic changes from population-level (i.e., cell count or proliferation marker) data. While we find that the relevant model parameter, $\beta$, is theoretically identifiable given a sufficiently large number of experimental observations, this identifiability is lost for imprecise cell-counts. Indeed, the difficulty in distinguishing between the functional form of the adaptation velocity (Fig 7), combined with the narrow time window in which the proliferation and death rates are distinguishable (Fig 4) suggests that heterogeneity may be indistinguishable from misspecification of other model terms. It is only if cell-level information (i.e., timings of proliferation and death events in the population) are available that we are, in theory, able to establish heterogeneity; although, the timescale of adaptation compared to the cell doubling time (less than 7 d compared to ~2–7 d for melanoma [6]) may arise as a practical limitation if individual cells are not observed to proliferate sufficiently many times during the adaptation phase.

A consequence of the non-identifiability of the parameter $\beta$ is that we cannot distinguish between a heterogeneous and a homogeneous model of continuous adaptation. Mathematically, this offers a practical benefit as it implies that population-level behaviours are well characterised by a simple, and in many cases analytically tractable, ODE model. All other model parameters: the on- and off-drug growth rates in each fully adapted state, and the adaptation velocity, are practically identifiable. In the supplementary material (S2 File), we explore a number of experimental designs in which various combinations of termination times are considered for a fixed total number of proliferation assays. Even if all experiments are terminated after 3 d, all relevant parameters remain identifiable; albeit estimates are drawn with reduced precision. The indistinguishability of the heterogeneous and homogeneous continuous transition model motivates us to explore the model selection question using an ODE-based homogeneous model. We are unable to distinguish the functionally correct adaptation velocity, however we do identify a class of models that manifest similar trajectories through phenotype space (Fig 7c).

The theoretical identifiability of heterogeneity from event-timing data using our stochastic formulation highlights two potential (and rarely considered) sources of potential misspecification in our model. First, that proliferation and death events are mutually exclusive: often it is only in a stochastic modelling framework that the two can be distinguished [46]. A more realistic (and correspondingly, further parameterised) model would consider individual and phenotype-dependent proliferation and death rates. Depending on the action of the drug and the metabolic cost of resistance, it may be appropriate for one of these rates to be phenotype-independent. For experiments initiated with a large number of cells, where cell competition may play a role, it may also be appropriate to consider a stochastic analogue of logistic, rather than exponential, growth. A second source of misspecification, the effects of which are, to the best of our knowledge, largely unknown in the context of population-level behaviour in IBMs, is that proliferation occurs according to a Markov process. Clearly, this is a strong assumption that, while routine in the mathematical literature, may be inappropriate. Given that adaptation occurs on a similar timescale as proliferation and that the proliferation rate varies, future application of established alternative models, for example those based on Erlang distributions [47], is not straightforward. Furthermore, any move away from a Markovian formulation in the IBM would render intractable the ODE, PDE, and CME formulations that we rely upon for inference.

Following the vast majority of the PDE literature, the heterogeneity in our model only manifests through random diffusive changes in phenotype [27]. Consequentially, all cells are statistically identical, and the phenotypic state of each cell is constantly evolving, even within a system that appears static at the population-level. It is only this formulation of heterogeneity that we find to be non-identifiable. A potentially more realistic model of heterogeneity is one that also considers inherent heterogeneity between individual cells; for example, variation in the growth rates of cells that are otherwise fully adapted (i.e., variation across cells where $x_i = 1$), or variation in the rate of adaptation for each cell. The question of identifiability of these population-level distributions from population-level statistics, such as cell count, remains open, although there is a fast-growing set of statistical tools that could be adapted to answer these questions [43,48]. Given the difficulty faced within our framework identifying variability in the instantaneous proliferation rate, we hypothesise that other sources of heterogeneity are likely to be non-identifiable (or at least, indistinguishable from measurement error or model misspecification) from population-level data.

Our final result is to demonstrate that the discrete-binary model of heterogeneous phenotypic adaptation is indistinguishable at the population level from an appropriately formulated continuous model. Our results do show very minor differences in high-order

behaviours (cell count variance), although we still expect both models to remain indistinguishable upon consideration of potential model misspecification and measurement noise. We do expect, however, the binary strategy to be distinguishable from marker data that correlate well with proliferation: the distribution of net growth rates in the binary model will always be bimodal throughout the adaptation phase, contrasting with the continuous transition we see both experimentally (Fig 1b), and in our model (Fig 6a). The equivalence we derive in Sect 3.5 also suggests at a hybrid discrete-continuous model that could be studied in future. Namely, a model in which drug-sensitive cells switch to an intermediate transition state with some drug-dependent propensity, in which the phenotypic state varies continuously until the cell reaches the drug-resistant state.

From a practical perspective, our work provides a statistical framework sufficient to characterise a population-level continuous phenotypic transition in response to a drug within relatively simple experiments. Key model parameters relating to the net growth rates and adaptation velocity were identifiable from an experimental design involving only eight standard 96-well plates (as few as two if images are taken as a time-series). To distinguish between a continuous phenotypic transition and the binary model of disparate sensitive and resistant phenotypes, or to establish the role of random diffusive phenotype changes (i.e., heterogeneity in the continuous framework), single-cell data are required. For example, marker data such as L1CAM (Fig 1b) are likely sufficient to distinguish between the binary and continuous models. In the continuous framework, however, heterogeneity can likely not be distinguished from either misspecification or marker noise. More sophisticated experiments, potentially based on microfluidics [49], may be necessary to accurately quantify heterogeneity in proliferation.

We establish the identifiability of reversible phenotype driven by both directed and random changes from commonly reported low-cell-count proliferation assay experiments. To achieve this, we develop a computationally efficient inference framework that captures potential information arising as intrinsic noise, without resorting to the study of a mean-field model subject to an additive Gaussian measurement process. That we find heterogeneity non-identifiable is significant to the mathematical modelling community, and implies that population-level behaviours (including, importantly, the response of systems to drugs and the design of adaptive therapies) are well characterised by homogeneous ODE models. For the experimental community, our methodology can be used to design and characterise experiments that probe continuous phenotypic adaptation in cancer.

## Supporting information

**S1 File. IBM/CME comparison.**
(PDF)

**S2 File. MCMC priors and results.**
(PDF)

**S3 File. CME for discrete model.**
(PDF)

**S4 File. Structural identifiability of heterogeneity.**
(PDF)

**S5 File. Large data set inference with noisy data.**
(PDF)

**S6 File. Inference with correlated data.**
(PDF)

## Acknowledgments

The authors thank Adriana Zanca for helpful discussions.

## Author contributions

**Conceptualization:** Alexander P. Browning, Rebecca M. Crossley, Chiara Villa, Philip K. Maini, Adrianne L. Jenner, Tyler Cassidy, Sara Hamis.

**Data curation:** Alexander P. Browning, Tyler Cassidy, Sara Hamis.

**Formal analysis:** Alexander P. Browning, Rebecca M. Crossley, Chiara Villa, Philip K. Maini, Adrianne L. Jenner, Tyler Cassidy, Sara Hamis.

**Funding acquisition:** Tyler Cassidy.

**Investigation:** Alexander P. Browning, Rebecca M. Crossley, Chiara Villa, Philip K. Maini, Adrianne L. Jenner, Tyler Cassidy, Sara Hamis.

**Methodology:** Alexander P. Browning, Rebecca M. Crossley, Chiara Villa, Philip K. Maini, Adrianne L. Jenner, Tyler Cassidy, Sara Hamis.

**Project administration:** Sara Hamis.

**Software:** Alexander P. Browning.

**Visualization:** Alexander P. Browning.

**Writing – original draft:** Alexander P. Browning.

**Writing – review & editing:** Alexander P. Browning, Rebecca M. Crossley, Chiara Villa, Philip K. Maini, Adrianne L. Jenner, Tyler Cassidy, Sara Hamis.

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
