## [Decision Letter · Decision Letter 0]

1 Apr 2025

PCOMPBIOL-D-25-00121

Identifiability of heterogeneous phenotype adaptation from low-cell-count experiments and a stochastic model

PLOS Computational Biology

Dear Dr. Browning,

Thank you for submitting your manuscript to PLOS Computational Biology. After careful consideration, we feel that it has merit but does not fully meet PLOS Computational Biology's publication criteria as it currently stands. Therefore, we invite you to submit a revised version of the manuscript that addresses the points raised during the review process.

Please submit your revised manuscript within 30 days Jun 01 2025 11:59PM. If you will need more time than this to complete your revisions, please reply to this message or contact the journal office at ploscompbiol@plos.org. Please include the following items when submitting your revised manuscript:

We look forward to receiving your revised manuscript.

Kind regards,

Guillermo Lorenzo

Academic Editor

PLOS Computational Biology

Mark Alber

Section Editor

PLOS Computational Biology

**Additional Editor Comments:**

Dear authors,

I would like to congratulate you on the excellent work presented in this manuscript, which has also been noted by the three reviewers. In general, we all agree that the manuscript is well-written, that the research is conducted with great rigor, and that the analysis of the results is insightful and comprehensive. Thus, we believe that this manuscript will be an important contribution to the field of mathematical oncology. Nevertheless, the reviewers have identified a few issues that need to be addressed before considering publication, which motivates my decision of "Minor Revision". I strongly recommend that you address all the points in the reviewers' reports in your response. In brief, their main comments require edits to address five points: (i) improve the contextualization of this work in the introduction and the discussion; (ii) clarify several fundamental concepts underlying the modeling framework; (iii) discuss whether conclusions hold for more complex treatment schedules; (iv) clarify several aspects of the model formulation and parameter inference approach; and (v) discuss how the results of this study could be leveraged to design experiments tailored for the identification of phenotypic heterogeneity.

We will be looking forward to seeing a revised version of your manuscript.

Best regards,

G. Lorenzo, Ph.D.

**Journal Requirements:**

At this stage, the following Authors/Authors require contributions: Alexander Browning, Rebecca M Crossley, Chiara Villa, Philip K Maini, Adrianne L Jenner, Tyler Cassidy, and Sara Hamis. Please ensure that the full contributions of each author are acknowledged in the "Add/Edit/Remove Authors" section of our submission form.

5) Please ensure that the funders and grant numbers match between the Financial Disclosure field and the Funding Information tab in your submission form. Note that the funders must be provided in the same order in both places as well. State what role the funders took in the study. If the funders had no role in your study, please state: "The funders had no role in study design, data collection and analysis, decision to publish, or preparation of the manuscript.".

**Reviewers' comments:**

Reviewer's Responses to Questions

**Comments to the Authors:**

Reviewer #1: This paper investigates the identifiability of key parameters governing heterogeneous phenotype adaptation in cancer, using a stochastic individual-based model (IBM) applied to low-cell-count proliferation assays. The paper is well written with a clear and well-structured mathematical framework.

The authors take a rigorous approach to the problem of identifiability, systematically analysing different experimental regimes and the conclusions are justified by the mathematical findings. Overall, I believe this paper is a strong contribution to the study of adaptive resistance in cancer, providing valuable insights into the identifiability of key model parameters. However, there are a few minor aspects that in my opinion need attention before publishing the paper.

1. The main aim of the paper is to "assess identifiability of key model parameters". However, the authors fail to introduce identifiability properly, e.g., to explain why is it important to consider it, or to reference previous literature satisfactorily. In particular I refer to the work by M.P. Saccomani, for example:

- G Bellu, MP Saccomani, S Audoly, L D’Angiò. DAISY: A new software tool to test global identifiability of biological and physiological systems. Computer methods and programs in biomedicine 88 (1), 52-61

- S Audoly, G Bellu, L D'Angio, MP Saccomani, C Cobelli. Global identifiability of nonlinear models of biological systems. IEEE Transactions on biomedical engineering 48 (1), 55-65

2. The study would benefit from a broader discussion of alternative sources of heterogeneity, practical experimental constraints, and potential ways to distinguish discrete and continuous adaptation in real-world data. I would suggest adding this aspects to the Discussion and Conclusion section.

3. The authors made the code available. However, the README file is of no help to understand how to run the code. Is there an order in which the files should be run? Is there a main file? What packages are needed? and so on. Please improve the code accessibility.

Finally some minor suggested changes are:

1. At pg 10 the authors write " ... all growth rate parameters are practically identifiable." What does that mean are they or not? I would remove the "practically".

2. At pg 11 there is a typo, soley should be solely.

3. I suggest using "Poisson" instead of "Po" when indicating a Poisson distribution.

4. At pg.14 it is unclear where the expression of v(x) comes from. Explain in more detail the function choice. Also, there is a typo and asgn, should be sgn or sign (if not, explain what you mean by that).

Reviewer #2: Please see the attachment.

Reviewer #3: In this study, Browning et al present a computational framework to infer how a population of cancer cells adapts to treatment from time-lapse data. Specifically, the framework examines the role of drug-induced, phenotypic switching of cells, as has recently been observed in several cancer settings. The authors perform simulation experiments to explore to what degree it is possible to infer the growth rates of cells in different phenotypic states, the rates of random and drug-induced switching, and whether or not changes in cell phenotype are continuous or discrete. In doing so, the authors address an important question regarding the calibration of mathematical models of resistance and the design of in vitro experiments that is timely given the growing evidence for the role non-genetic processes in cancer drug resistance evolution.

This is a strong and well written manuscript which will be of great interest to readers of this journal. It is essentially ready for publication, except for three points which I would ask the authors to address to clarify some of the language in the paper. I also have few additional minor suggestions/typos that can be found at the end of my response. Thank you for inviting me to contribute to this manuscript as a reviewer and congratulations to the authors on a great piece of work!

Major comments:

- “Adaptive Resistance”. I have not heard the term “adaptive resistance” before. I’m happy to accept my ignorance here, but I find this particular term a bit confusing because of the strong history and meaning that the term “adaptation” has in classical evolutionary theory. Specifically, what worries me about “adaptive resistance” is that it confounds whether the adaptation is happening on the individual level (as in scenario you are studying) or on the population level (as assumed in classical evolutionary theory, which can be brought about by genetic changes (Darwinian Model) and/or phenotypic changes (Lamarckian Model)). There isn’t great consensus yet around terminology for the drug-induced and heritable phenotypic changes we are seeing in cancer, but in order to better connect with the existing literature I would suggest you to mention some of the terms others have used (specifically the papers you are citing). e.g. “phenotypic switching”, “epigenetic plasticity”, “drug-induced resistance”, or “non-genetic resistance”. And if you don’t feel strongly, maybe even consider adopting one of those terms instead of introducing a new one.

- Similarly, I would ask you to change the way in which you use the term “heterogeneity”. From what I can tell you use it to describe two related, but distinct phenomena: i) Whether cells exist in 2 discrete states (sensitive and resistant) or in a continuum of states along a sensitive-resistant axis, and ii) whether switching of cells between states with different drug sensitives is random (diffusive) or directed (advective/drug-induced). Firstly, I would argue that since all your models include sensitive and resistant cells in some form they model a “heterogeneous” cell population (at least in the way most cancer biologist would think about it). So, I’m not sure that “heterogenity” is a good word to use for either phenomenon. Regarding Point i), evolutionary biologists use the terms “gradualism” (many small changes) and “saltationism” (few, big changes; I’ve also heard “step-wise evolution”) to distinguish between these scenarios. Perhaps one of those would be clearer? Regarding Point ii) I think this distinction would be worth explicitly highlighting. The question whether phenotypic adaptation happens randomly (drift) or is drug-induced (advection) is one of the key questions in cancer biology at the moment. I think a discussion of this point and how your framework can help to answer this question would be a great addition to your manuscript. See my comment about Page 10 for more details.

- Choice of experimental setup: You conclude that models that assume discrete or continuous phenotypic transitions can not be distinguished, and we’re justified in using the simpler 2-state ODE models. That’s great news, but I worry that to some extent that this might be confounded to the experimental setup you are studying here. Specifically, the way I understand it all your experiments happen in a constant drug environment (either all on or all off). You present some intermittent data initially, but then partition it into either the on or the off phase. That means you are comparing these models in how they describe the switching from one end of the sensitive-resistant axis to the other (either in a gradual or discrete fashion). We are essentially seeing these cells “roll down a hill” in phenotype space and I think it makes sense that there is little difference in this scenario whether this process is happening in a continuous or step-wise fashion. However, I suspect this might be different under more complex treatment schedules, where it can matter that we have cells half-way between the sensitive and resistant state when we switch the drug concentration. I have added a few papers that demonstrate this. I would be interested in hearing your thoughts about this question. Unless I have misunderstood the implications of your work, I would suggest to add a discussion of this point in the discussion. If you have time I would also be curious to see if your conclusions in Section 3.5 might change if we subjected these cells to intermittent schedules?

-- Gunnarsson, E. B., Magnússon, B. V. & Foo, J. Optimal dosing of anti-cancer treatment under drug-induced plasticity. arXiv (2024) doi:10.48550/arxiv.2412.16391.

-- Corigliano, M., Bernardo, A. D., Lagomarsino, M. C. & Pompei, S. Optimal treatment for drug-induced cancer persisters involves release periods and intermediate drug doses. bioRxiv 2024.11.29.626082 (2024) doi:10.1101/2024.11.29.626082.

-- Smalley, I. et al. Leveraging transcriptional dynamics to improve BRAF inhibitor responses in melanoma. EBioMedicine 48, 178–190 (2019).

-- Raatz, M. & Traulsen, A. Promoting extinction or minimizing growth? The impact of treatment on trait trajectories in evolving populations. Evolution 77, 1408–1421 (2023).

Minor comments:

Fig 1c: could you provide a bit more explanation of what the schematic is showing?

Page 3: “intermediate period between resensitisation and addiction” → are you referring to “drug addiction” here? If so, I’d suggest explaining what this means and how it relates to this study. I’m also not sure I see evidence for addiction (higher growth rate under drug vs no drug) in your data. Might be simpler and clearer to just say something like “during the re-sensitisation period between days 7-14, when drug was taken off”?

Page 4: “untreated and treated states”; maybe use a different word than “states” to clarify that this to the state of the environment, and NOT the cells.

Page 5:

- What is x?

- What are gamma1-4?

- Could you provide a 1 sentence motivation/explanation for the functional form for lambda in Eq 2? Perhaps, you could add a small panel to fig 1 illustrating the assumed relationship between x and lambda.

- “Provided that the growth rate is monotonic in x, the functional form of λ is arbitrary since we could, in theory, rescale the phenotypic space in Eq. (1) and thus equivalently the functional form of v”. I’m not an expert in this but this feels like a very strong statement. What if lambda is non-linear? Presumably you’d need more than a linear scaling to adjust v to that?

Figure 2:

a) “addicted” → should this say “resistant”?

e & f) It took me a moment to understand what the difference between the two panels is since both include intermittent treatment. could you add a label saying (drug on) and (drug off) to the titles, since this is really what discriminates the two panels?

Page 8: “The advantage of this approach, compared to a more standard approach that considers an average cell count subject to additive Gaussian noise, is that we account directly for the stochasticity intrinsic to the proliferation death process.” → Nice! I think that is a really interesting and useful point.

Page 9:

- “kth cell count” → do you mean count of cells with phenotype k?

- “assuming that measurements are taken at the termination of an experiment and not as a time-series” → I got a bit confused when I got to this point. With the time-series images and data you show in fig 1 I was expecting you to use the time-series data. But if I’m understanding this correctly, you are only using a single data point? If that is correct, then I would suggest that you make this assumption clearer in your introduction and discussion too. I don’t think it’s necessarily a limitation, since it’s impressive and useful to know what we can do with a single data point (also thinking about potential in vivo translation/working with more complex experimental setups). But I think it would be important to highlight this more clearly. I also wonder if you were to use the time series data you could address some of the limitations that you identify, since beta will also manifest itself in the time correlation structure?

Page 10:

- “one-sided identifiable”: I like this term and your thoughtful analysis here. With that said, I wonder if you’re being overly pessimistic? Yes, you can’t set a lower bound, but isn’t the upper bound more interesting anyhow in answering the question of how much random drift contributes to the observed dynamics? By being able to say it’s less than x and comparing that bound to nu we can deduce how much each of the processes are driving the adaptation.

- “Thus, we expect, from cell count data alone, models with phenotypic heterogenity to be indistinguishable from models with deterministic adaptation (the β = 0 scenario)” → From what I understand what you mean to say here is that you can not distinguish between a scenario in which adaptation is driven purely by drug-driven phenotype switching, and a scenario in which there is both drug-driven switching (the advection term) and random switching (diffusion term; cells can move both up and down the phenotype axis, at a rate that’s independent of drug). If this interpretation is correct, then I wonder if this really is a helpful question to ask? Wouldn’t one always expect there to be a certain degree of “noise” in biological processes? When, in practice, would we expect a process to be full deterministic? If I may, I would argue that would be more helpful to know is the relative balance between noise and drug-induced switching. Specifically, in the setting of drug resistance evolution, the key question would be: is adaptation happening through random switching, or is it drug-induced, i.e. how strong is the contribution of noise/drift relative to the movement caused by the drug-induced switching/advection term? Could you comment on this, and clarify in the text here what exactly you mean with your statement and what it’s practical implications are? Related to my major point, may I also suggest to use a different word than “heterogeneity”. I’d argue that you have heterogeneity (sensitive and resistant cell types) even if beta=0. If you have time, it might also be interesting to explore how well your approach can distinguish the two evolutionary scenarios I’ve outlined above (drug-induced (advective) vs random (diffusive) switching). This would be of great practical interest and would strengthen the potential impact of your work.

Figure 3:

- “model predicted mean cell count” → which model is this referring to? Could yo reference the equation here for clarity?

- c & d? From what I understand your inference method only uses the data from a single time point (“end point”) for inference. So while all the data in a and b are available in principle, your estimates in c and d only use one of the time points. Which time point is it? And, could you also comment on what effect the choice of time point has on our ability to infer parameters? Is there a optimal length of time one should conduct this experiment over?

Figure 4: missing closing bracket after “maximum proliferation rate”

Page 11:

“most obvious consequence of the β = 0 parameter regime is that proliferation and death cannot occur simultaneously” → why is that? I’m not sure I’m following. Sorry if I’m missing something. Could you provide more detail here?

“proliferation-death transition” → Where is that happening? Could you indicate that in fig 4?

“soley” → typo

“Under the current model formulation, in which heterogeneity is driven soley by diffusion through the phenotypic space, it is only in the regime where β > 0 that we will ever see proliferation and death events occur simultaneously.” → Related to my major points, I think “heterogeneity” is a confusing word to use here, since you have the possibility of different cell states even without beta. And related to my question above, it is not clear to me why we can’t have proliferation and death events at the same time.

Page 12:

- \delta t = 0.035 → Can you add units? This will be useful to understand what your bin size means practically and how it compares to standard imaging frequencies.

- “previousl in Fig 5a” → did you mean to reference a different figure here?

- “Clearly, heterogeneity is now identifiable; estimates of β can be drawn precisely.” → see major comment around your use of the word “heterogeneity”.

Page 13:

- M_i → What’s the subscript i referring to?

- Figure 6: It looks like the value you’ve assumed for sigma is larger than that of beta. It’s hard to compare these two values directly, but could it be that the level of noise you’re assuming here in the markers is higher than the noise that is generated by the intrinsic stochasticity in the cell dynamics? If so, could that be the reason why we cannot identify beta? What happens if sigma = 0?

- Isn’t beta one0-sided identifiable again?

Page 14:

- Equation 16: what’s “asign(x)”? Could you define that function? It looks like it might be a typo and is meant to read a*sign(x); even so I think it would be helpful to define what sign(x) is.

- a=c=d=0: why are you using this as a test case? Is there particular practical or theoretical motivation? It seems odd to introduce a model with 4 parameters, but then discard 3 of them in the application.

Figure 7: what is x(t)? Is that the average phenotype?

Page 15:

- “we compare the likelihood at the MLE (equivalently, the MAP) for a = 0, denoted by theta[a]” → this makes it sound like you’re denoting the likelihood by theta[a], when from what I understand you theta is meant to denote the parameter set?

- “we compare the identified adaptation velocities … to the full model” → what’s the “full” model? Is that the ground truth model?

Section 3.5.

Related to my major comment: how much do the conclusions drawn in this section depend on your assumption that lambda is linearly correlated with x, and that we only have data from experiments under constant treatment conditions (e.g. all on or all off)?

Page 17:

- “may arise as a practical limitation that prevents heterogeneity from becoming practically identifiable” → related to my major comments and my comment on page 10: isn’t it useful to be able to put an upper bound on how much noise is contributing? If it’s fairly small that would give an indication that drug-induced adaptation plays a significant role in this process.

- “we cannot distinguish be- tween a heterogeneous and a homogeneous model” → related to my major comment: I’d strongly argue to use different terminology. Both models include heterogeneity in that there are cells in different cell state. It’s just the number of these states that is different (gradualism vs saltatism).

- “sources of model misspecification” → by whom? It’s unclear if this is referring to the field as a whole or to your work.

- “large-cell-count experiments” → large in what way? Long time? Large seeding numbers? Many replicates?

- “Given that adaptation occurs on a similar timescale as proliferation, what an appropriate time-to-proliferation distribution would be is entirely unclear.” → Love it! If you are interested in pursuing this you might enjoy the CFSE tagging literature. See e.g. these two papers as intros:

-- Miao, H., Jin, X., Perelson, A. S. & Wu, H. Evaluation of Multitype Mathematical Models for CFSE-Labeling Experiment Data. Bulletin of Mathematical Biology 74, 300–326 (2012).

-- Boer, R. J. D. & Perelson, A. S. Estimating division and death rates from CFSE data. Journal of Computational and Applied Mathematics 184, 140–164 (2005).

Page 18:

- “where individual cells have a unique growth rate when both fully sensitive and fully adapted” → wait, I though each x corresponds to a unique lambda so that each state has a unique rate associated with it?

- “ significant to the mathematical modelling community, and implies that population-level behaviours (including, importantly, the response of systems to drugs and the design of adaptive therapies) are well characterised by homogeneous ODE models” → I find this a very helpful way of putting your results into context. And I find it a comforting result to read. With that said, as mentioned in my major comments, I worry that this equivalence might start to break down in the case of intermittent schedules where the cells don’t get time to full transition into the other state before the dose is changed again. In this case it feels like the two types of approaches might give different results.

Supplementary material

Figure S2: how are you combining data from different time points when you are doing the inference?

Figure S3 (and S5): Can you add a label to indicate that a) and b) are sensitive and resistant cells respectively?

Figure S4: “re reproduce the results” → “we repeat the (computational) experiment”? I got confused by “reproduce” and thought you were showing the same data again.

Section S5:

- Could you provide some more justification/motivation for why you chose this error models and that particular set of parameters? What is the underlying biological motivation?

- Isn’t beta one-sided identifiable again?

**Have the authors made all data and (if applicable) computational code underlying the findings in their manuscript fully available?**

Reviewer #1: Yes

Reviewer #2: Yes

Reviewer #3: Yes

PLOS authors have the option to publish the peer review history of their article (what does this mean?). If published, this will include your full peer review and any attached files.

Reviewer #1: No

Reviewer #2: **Yes: **Simon Syga

Reviewer #3: No

**Figure resubmission:**
---

## [Decision Letter · Decision Letter 1]

4 Jun 2025

Dear Dr Browning,

We are pleased to inform you that your manuscript 'Identifiability of phenotypic adaptation from low-cell-count experiments and a stochastic model' has been provisionally accepted for publication in PLOS Computational Biology.

Best regards,

Guillermo Lorenzo

Academic Editor

PLOS Computational Biology

Mark Alber

Section Editor

PLOS Computational Biology

Dear authors,

many thanks for addressing all the points raised by the reviewers, who have all now agreed that the work is ready for publication.

I would like to join them in congratulating you and your colleagues on the work carried out in this manuscript. I look forward to seeing your future work in the field.

Best regards,

GL

Reviewer's Responses to Questions

**Comments to the Authors:**

Reviewer #1: The authors addressed all of my previous comments.

Reviewer #2: The authors did a great job addressing all comments. I believe that this manuscript will be a valuable contribution to the community and I recommend publication.

Reviewer #3: Thank you for your thoughtful and thorough reply. You have addressed all my concerns (and often times gone above and beyond!). I was surprised to see that the intermittent schedules don't aid with identifiability. Thank you for doing this additional work - that's very helpful to know. Congratulations on a fantastic piece of work!

**Have the authors made all data and (if applicable) computational code underlying the findings in their manuscript fully available?**

Reviewer #1: Yes

Reviewer #2: Yes

Reviewer #3: Yes

PLOS authors have the option to publish the peer review history of their article (what does this mean?). If published, this will include your full peer review and any attached files.

Reviewer #1: No

Reviewer #2: **Yes: **Simon Syga

Reviewer #3: No

---

## [Editor Report · Acceptance letter]

PCOMPBIOL-D-25-00121R1

Identifiability of phenotypic adaptation from low-cell-count experiments and a stochastic model

Dear Dr Browning,

I am pleased to inform you that your manuscript has been formally accepted for publication in PLOS Computational Biology. Your manuscript is now with our production department and you will be notified of the publication date in due course.

With kind regards,

Anita Estes
